# Ultrasound Elastography in Children

**DOI:** 10.3390/children10081296

**Published:** 2023-07-27

**Authors:** Mirjam Močnik, Nataša Marčun Varda

**Affiliations:** 1Department of Paediatrics, University Medical Centre Maribor, 2000 Maribor, Slovenia; natasa.marcunvarda@siol.net; 2Faculty of Medicine, University of Maribor, 2000 Maribor, Slovenia

**Keywords:** ultrasound, elastography, children, indications

## Abstract

Ultrasound elastography is a novel ultrasound technique, being extensively researched in children in the last decade. It measures tissue elasticity with the observation of tissue response after an external stimulus. From research to clinical practice, ultrasound elastography has evolved significantly in liver fibrosis evaluation in children; however, several other applications of the technique are available in both clinical practice and research environments. Practically, in children any organ can be assessed, including the brain in early ages, along with muscle and connective tissue elasticity evaluation, spleen, kidney, skin, lymphatic tissue, etc. The elastography method, age, body mass index and technical points should be considered when performing ultrasound elastography. This review highlights its vast potential as a diagnostic tool in the pediatric population, where ultrasound is a dominant imaging tool; however, the indications and exam protocol for its clinical use in several fields in pediatrics have yet to be elucidated.

## 1. Introduction

Ultrasound is one of the most important diagnostic tools for children, as it is easily accessible, does not require sedation, can be repeated several times and can be used for countless indications. Not only does it display an image of the organ, additional applications of the probes enable many additional investigations such as vascular flow using Doppler techniques, blood supply and tissue elasticity. The measurement of the latter has been intensively researched in recent years, and the method we use to perform it is called ultrasound elastography (UE) [1,2,3,4]. The use of UE opens up a new spectrum of ultrasound use and could be easily introduced into standard ultrasound examination if required. In recent years, the use of UE in various fields has also been intensively researched in children [1].

The aim of this review was to present the status of UE use in children as a diagnostic or research tool and to encourage its continuous use in clinical practice and further research. A similar review was already published and could be studied in conjunction with our study [4]; however, our study extends to applications to other tissues not yet presented (pancreas, bladder, salivary glands, skin) and in the context of cardiovascular risk assessment, including new studies not yet presented.

We searched for studies reporting on UE in children in PubMed, Web of Science and Google Scholar databases. The PubMed search in June 2023 using “ultrasound elastography in children” yielded 1207 results on several topics in pediatrics, presented in Figure 1. The results were next filtered according to their significance, methodology and examined organ, presenting several indications for use of UE, as presented in this review.

## 2. Methodological Aspects of Ultrasound Elastography in Children

Elasticity is defined as the tissue’s tendency to resist deformation or to recover its shape after the removal of an external force. Different tissue elasticity can be the result of physiological or pathological factors [2]. To compare the elasticity of the tissues, the displacement speed with which the region of interest responds to the applied stimulus is used. Tissue movement is then detected using a variety of manufacturer-dependent methods, including radiofrequency, echo-based tracking, Doppler processing or combinations of these [2,3].

The measurement starts with a good visualization of the target organ with classical imaging. Next, the appropriate software is switched on at the desired location within the organ and points at it with the indicator, as presented in Figure 2. Several elasticity areas within the organ are measured and the mean or median value of measurement is provided using speed (m/s) and pressure (kPa) modules, both describing tissue’s elastic properties. The quality of the measurement is ensured by the manufacturer’s instructions.

UE is thus a method by which an external force is applied with an ultrasound probe and then the response of the tissue is observed, leading to tissue elasticity assessment. Depending on the direction of the ultrasonic waves, we distinguish strain (particles travel parallel to the waves) and shear forces (particles travel perpendicular to the waves). External stimuli vary from manual pressure to vibrations and acoustic pulses [2,3].

The review includes only ultrasound elastography; however, in recent years magnetic resonance elastography has also been gaining attention in several fields in pediatrics but was excluded in this review to retain coherence.

The measurement of tissue elasticity with the observation of organ deformity sounds simple; however, the reality is quite complicated. UE can be performed in different ways, shown in Table 1 [3,5], using different ultrasound machines and probes that are not interchangeable. Even the acquisition of measurements using a convex compared to a linear transducer leads to higher variability [6].

All shear wave imaging techniques have been introduced mainly for liver elastography, showing great feasibility in children [7,8]. Point shear wave and 2D shear wave elastography were also used for the assessment of other organs [9]. However, normal liver stiffness values in children differ according to the technique used and therefore they cannot be interchangeable [7]. Also, the precision and the diagnostic accuracy of all techniques must still be determined. Transient elastography was developed specifically for liver fibrosis evaluation, but recently a study demonstrated that it does not have acceptable precision in healthy children, showing the need for refinements to the test protocol [10].

When performing UE in children, several other aspects need to be considered, such as age, body mass index and technical points. Age is an important factor influencing anatomical and physiological differences across different age groups. In addition, younger children are less likely to cooperate and may not have the patience for the required number of measurements [9]. Furthermore, stiffness itself was related to age and body mass index [11,12]. The latter might also affect the examination with positive correlation to data variability [13]. The state of the patient is also of great importance, as there was an impact on liver stiffness measurement while fasted and rested vs. after exercise and during intercurrent illness [9,11].

Current recommendations include breath-hold conditions, which makes UE impossible in babies and difficult in younger children. Interestingly, 2D shear wave elastography with a free-breathing method yielded results similar to the breath-hold conditions with a lower time requirement, allowing the UE to be suitable for infants and less cooperative children not capable of holding their breath [14].

Several technical points, such as difference in location for measurement, the selection of the probe or the position of the patient, also need to be considered [9]. In addition, the numbers of acquisition differ from the general recommendation of ten acquisitions regardless of elastography technique. Comparing the number of acquisitions in all shear wave elastography techniques showed no significant difference between three and ten acquisitions. To achieve less than 20% liver stiffness measurement variation in children, in point shear wave and transient elastography four acquisitions were needed and in 2D shear wave elastography six acquisitions were needed, contradicting current recommendations [15].

## 3. Liver Ultrasound Elastography in Children

Liver UE is most commonly performed when using elastography in children. Shear wave elastography was found to be a feasible method to measure liver stiffness in the healthy school age population with the establishment of a normal range of liver elastic modules [16]. Specific diseases, different from those in adults, are more likely to cause chronic liver disease in children, such as cystic fibrosis-associated liver disease, metabolic liver diseases, Gaucher’s disease, Wilson’s disease, ɑ1-antitrypsin deficiency, glycogen storage disease, biliary disease, steatohepatitis, corrective surgery of congenital heart diseases, etc., where liver UE is indicated for liver fibrosis assessment and follow-up [9]. Application of UE instead of repeated liver biopsies in patients is a much simpler and less invasive method providing an opportunity for preventing unnecessary liver biopsies [17].

UE showed the ability to diagnose cirrhosis, distinguish healthy from fibrotic liver tissue and also showed consistent liver stiffness values in children without liver disease across studies [18]. Furthermore, transient elastography and 2D shear wave elastography showed a good correlation to the degree of liver fibrosis when compared to histological samples [19,20,21], despite questioning on the precision of transient elastography [10]. Another study also achieved good liver fibrosis detection; however, it was not able to differentiate fibrosis stages [22].

Several other studies have confirmed 2D shear wave elastography to have good sensitivity and specificity and to be reliable in assessing liver fibrosis in children [23,24]. On the other hand, liver inflammation or higher transaminase levels were found to be potential confounders for liver fibrosis staging when performing UE [25,26]. Despite this, UE is being increasingly recognized as a promising technology for the diagnosis of advanced fibrosis in children with hepatitis B [27,28]. Similarly, shear wave elastography correlated significantly with liver biopsy histological evaluations in children with hepatitis C, differentiating between no or mild fibrosis and significant fibrosis [29].

Cystic fibrosis-related liver disease is another indication for UE, where several techniques were found to be reliable methods to evaluate liver fibrosis and were proposed for the follow-up of patients with cystic fibrosis [22,30].

Non-invasive examinations are also necessary to monitor liver fibrosis in children with biliary atresia after the Kasai procedure, where UE showed good accuracy and was proposed to yield more accurate diagnoses of liver fibrosis when combined with serologic markers [31,32] or to possibly assist in selecting appropriate biopsy location [33].

Several other conditions were also researched in regard to liver UE, which was found to be useful and could have a clinical impact in diagnosis or follow-up of, for example, non-alcoholic fatty liver disease [34], short bowel syndrome [35], portal hypertension [36], posttransplant graft fibrosis [37,38], autosomal recessive polycystic kidney disease [39] and Wilson’s disease, where decrements of elastography parameters were also noted during specific treatment [40,41]. Similarly, liver stiffness was evaluated in children with juvenile idiopathic arthritis receiving methotrexate to detect possible liver toxicity effects; however, no increase in liver tissue stiffness was noted in these patients [42].

## 4. Spleen Ultrasound Elastography in Children

The spleen can also easily be examined with UE and was found to be stiffer than liver and kidneys in healthy children and showed age-related changes in one study [43] that were not proven in others [44,45]. Normal values for spleen stiffness were obtained with no difference in the mean values using two types of ultrasound transducers [6] or when using different probe positions [45].

Spleen stiffness measured by transient elastography was significantly higher in children with splenomegaly, in patients with varices and in patients with a history of variceal hemorrhage [46]. The degree of spleen stiffness was predictive of the severity of portal hypertension in patients with biliary atresia after the Kasai procedure [47]. Also, spleen stiffness measured with transient elastography decreased after liver transplantation but remained elevated compared with healthy children [46]. Spleen stiffness measurement was found to be useful as a non-invasive screening tool for children with portal hypertension to stratify the risk of having clinically significant varices [48], with its diagnostic yield increased when combined with liver stiffness assessment [49]. Spleen stiffness was also higher in children after living donor liver transplantation with venous complications and could therefore provide a useful screening tool for detecting venous complications in this cohort of pediatric patients [50].

## 5. Pancreas Ultrasound Elastography in Children

Pancreas UE is feasible in healthy children using strain or shear wave elastography [51,52], showing age and body mass impacts; the former was most prominent with the transition from childhood to adolescence [52,53].

Strain imaging showed increased strain ratios in children with type 1 diabetes mellitus, presenting an additional method for early detection, long-term screening and follow-up in children with type 1 diabetes mellitus [54]. Similarly, shear wave imaging changed significantly with duration of type 1 diabetes mellitus, particularly in those with diabetic nephropathy and neuropathy [55], but not all studies confirmed that [56].

The pancreas can be severely affected by cystic fibrosis in children, where increased shear wave values demonstrated a potential of UE for early screening in the pancreas and liver before clinical signs appear [57]. Confusingly, two other studies found significantly lower pancreatic UE values in children with cystic fibrosis compared to healthy controls [58,59], showing troublesome inconsistency.

## 6. Kidney Ultrasound Elastography in Children

Kidney UE showed age-related changes in healthy pediatric volunteers, but only in those younger than 5 years of age according to one study [43] and across all age groups in another with the addition of weight and probe dependency [60].

Kidney stiffness in kidney disease with preserved kidney function varies according to diagnosis. Children with acute glomerulonephritis had higher mean elastography values than healthy controls [61]. Similarly, renal cortical stiffness was higher in overweight children [62,63]. On the contrary, children with type 1 diabetes mellitus and normoalbuminuria had similar kidney stiffness values compared to healthy controls [56]. Furthermore, children with severely damaged kidneys due to vesicoureteral reflux had lower UE results [64], similar to the kidneys, affected by ureteropelvic junction obstruction [65].

In children with nephropathy, compared to healthy children, UE seems to be inconsistent due to heterogeneous diseases affecting kidney function, expressing the need for studies with larger sample sizes [63,66]. Regardless of the cause, the progress of chronic kidney disease was associated with higher kidney stiffness in children [67]. Interestingly, correlations between elasticity and kidney fibrosis were not demonstrated [68]. Similarly, there was no difference between renal units with or without scar formation using UE [69], which may be due to the kidney’s complex structure and variations in blood perfusion and the glomerular filtration rate of the kidneys, currently limiting the use of UE as a diagnostic tool in kidneys [70].

## 7. Bladder Ultrasound Elastography in Children

UE in the evaluation of neurogenic bladders with measurements of the bladder wall, compliance and capacity showed promising results in an adult population [71]; however, the same was not confirmed in children in the non-distended bladder [72]. At present, the data on bladder wall evaluation with UE in children are lacking but promising with correlation between shear wave imaging and bladder thickness in children with acute cystitis [73].

## 8. Thyroid Ultrasound Elastography in Children

Thyroid UE is feasible in healthy children, again demonstrating an association with age and body mass index in some studies [74] but not in all [75].

One of the most important indications of UE in adult populations is malignancy differentiation [76], which might be applicable in thyroid nodule evaluation in children [77]. High elasticity of a nodule was associated with a low risk of thyroid cancer [78]. Another use of UE is in making decisions about fine needle aspiration cytology [79], which remains the gold standard for thyroid nodule assessment. 

In children with Hashimoto’s thyroiditis, shear wave elastography showed increased values compared to healthy controls and increased significantly with the increment of thyroiditis stage [80,81]. The same was true for Grave’s disease; however, UE was not successful in differentiation between these types of thyroid inflammation [82]. The strain index of thyroid UE was also higher in children with congenital hypothyroidism compared to healthy controls [83].

## 9. Lymphatic Tissue Ultrasound Elastography in Children

UE was found to be a useful tool in assessing cervical lymph nodes in detecting pediatric malignancy, where UE achieved sensitivity of 100% and specificity of 85.7% with an overall accuracy of 90.6% in the differentiation between malignant and benign entities. Despite these favorable results, UE cannot replace surgical biopsy [84]. On the other hand, the strain index showed less promise in differentiation of benign and malignant cervical lymph nodes with the sensitivity, specificity and accuracy of 71.6%, 76.5% and 75.0%, respectively [85].

Lymphatic tissue evaluation extends beyond cervical lymph nodes, at least in part, with higher shear wave elastography values found in the patients with acute tonsillitis [86], encouraging further use of UE in other lymphatic tissues.

## 10. Muscle and Connective Tissue Ultrasound Elastography in Children

Diagnosis of muscular diseases in children is challenging and the gold standard methods are often invasive or need good cooperation. Thus, UE may aid in both the diagnosis and monitoring of muscle diseases with some specific considerations [87]. An increase in muscle stiffness with age was observed in most pediatric studies, with some minor deviations [87,88]. Most importantly, muscle stretching, effort and exercise significantly influence muscle stiffness [89,90]. Exercise increases muscle stiffness and the change is greater in younger children [90].

Muscle UE was feasible in healthy children, demonstrating lower muscle elasticity at rest when compared to adults [91]. Tissue elasticity measurement was also feasible in smaller muscles and joints, such as the masseter muscle and temporomandibular joint [92,93].

One of the increasing pathologies in pediatric populations is cerebral palsy, where a significant burden is present among patients and their families considering the patient’s ambulation. This area is being substantially researched with UE, showing passive muscle stiffness in patients with higher shear wave speed in the more-affected limb [94,95,96]. The difference in passive muscle properties was noted in children with cerebral palsy after botulinum therapy, indicating its role in follow-up of the therapy [97].

Another devastating disease in terms of patient movement is Duchenne muscular dystrophy, where UE muscle stiffness was moderately higher compared to healthy controls across several muscle groups, providing an additional monitoring tool [98,99].

UE was considered an additional tool for diagnosis of benign acute childhood myositis [100]; however, it did not accurately detect active myositis in patients with juvenile idiopathic inflammatory myopathy [101]. In addition, UE was also used for monitoring after surgical treatment of developmental hip dysplasia [102] and in evaluation and prevention of sports injuries [103].

Another rising medical issue in children and adolescents is scoliosis, which has also been researched in terms of UE of intercostal muscles and intervertebral discs, which was feasible in this cohort of patients [104,105]. Some studies reported significantly higher shear wave speed in scoliosis patients compared with healthy controls [106,107], making UE a potential biomechanical marker of idiopathic scoliosis [107].

## 11. Parotid and Submandibular Ultrasound Elastography in Children

UE of parotid and submandibular glands is feasible in children and research has provided some baseline evaluations of parenchyma in children; however, they are inconclusive—some show correlation of salivary glands with age [108,109] and body mass index [109], while others do not show such an association while also showing no difference between boys and girls [110]. UE seems promising for tumors in salivary glands and in diagnosis and follow-up for patients with recurrent parotitis; however, these results are limited to adult populations [111,112].

## 12. Brain and Nerve Ultrasound Elastography in Children

Pediatric patients are uniquely accessible for brain imaging with ultrasound in the earliest stage of life. In the neonatal brain, greater stiffness was observed in term infants compared to preterm ones, supposedly due to the greater degree of myelination present in term infants [113], showing a possible application for UE as a prognostic marker of poor neurodevelopmental outcome in preterm infants. In addition, current data show that elastography can be helpful in detecting brain injury and monitoring changes in hydrocephalus [113,114]. Infants with large intraparenchymal hemorrhage had increased white matter stiffness [115]. Adequate education on safety aspects of ultrasonography is of special importance for UE of the delicate infant brain, emphasizing limiting ultrasound exposure and balancing the potential bioeffects with the benefits of the obtained information [116].

Peripheral nerves are, in contrast to the brain, also accessible later in a child’s life where UE demonstrated promising results in detecting subclinical diabetic peripheral neuropathy in type 1 diabetic adolescents, however, with low reliability [117].

## 13. Scrotum and Testicle Ultrasound Elastography in Children

Testes pathology is also frequent in early ages with the testes being easily accessible with ultrasound. Normal changes and ranges for pediatric testicular volume and shear wave elasticity were determined with testes volume increasing with age along with decreasing testis stiffness [118]. Using strain imaging in boys with testicular torsion, testicular space-occupying lesions and epididymal lesions were shown to increase stiffness [119,120].

In some specific cases, UE could also aid in differentiation between undescended testes and reactive lymph nodes but conventional ultrasound mostly provides sufficient information [121]. Undescended testes exhibited smaller volume and increasing stiffness [122]. Interestingly, testes with non-communicating hydrocele or with varicocele in infants were significantly stiffer than normal ones [123,124]. Stiffness values were also significantly higher in testes with microlithiasis as compared to the control group, showing UE’s role in follow-up examinations [125].

In children with proven testicular involvement of hematological malignancies, increased testicular stiffness was demonstrated, indicating its potential role in monitoring children with hematological malignancy [126].

## 14. Skin Ultrasound Elastography in Children

Our biggest organ is the skin and it is also easily accessible with ultrasound; however, ultrasound use is limited to certain pathologies, where additional information is provided along with inspection. One of them is localized scleroderma, where UE showed that lesions with scleroderma in children had significantly higher stiffness values than controls, demonstrating UE’s feasibility to differentiate between normal and affected skin [127]. Similarly, UE demonstrated differences between pediatric burn scars and normal skin, presenting a potential tool to objectify scars or possibly to detect scar change over time or in response to treatment [128]. UE also demonstrated increasing temporal skin thickness and stiffness after cochlear implant surgery [129].

## 15. Ultrasound Elastography as an Early Screening Method of Cardiovascular Risk

Cardiovascular diseases are the top causes of morbidity and mortality worldwide [130]. Several cardiovascular risk factors are clinically silent until a cardiovascular event happens with devastating consequences. A universal, non-invasive and reliable screening tool to detect individuals at risk is still needed. In adults, intravascular strain UE, presented as a method for measuring the local elastic properties of coronary or carotid atherosclerotic plaques, showed promising results for the detection of vulnerable plaques [131].

UE was also used in children with hypertension or chronic kidney disease to assess its usability in detecting early changes concerning cardiovascular risk evaluation, where in both groups of patients liver stiffness parameters were increased and aggravated by obesity. The latter was also decisive in kidney UE and increased in children with co-existing chronic kidney disease, indicating a possible negative effect of clustering cardiovascular risk factors along with the feasibility of UE to detect the increased stiffness of target organs [63]. Thus, these studies may possibly provide foundations to detect early cardiovascular risk and early target organ changes; however, further research is needed to confirm that, to develop exam protocols and to determine the grade of stiffness with increased cardiovascular risk.

## 16. Advantages and Limitations of Ultrasound Elastography and the Summarization of Its Clinical Use

Despite its advantages, the use of ultrasound elastography in clinical practice is limited. Its advantages and limitations are presented in Figure 3 [1,2,3,4]. The main advantages in pediatrics include its accessibility, non-invasiveness (lack of ionizing radiation, no need for sedation), repeatability and ease of its clinical introduction for measuring the stiffness of various tissues [1]. Imaging is generally in real time, the exposures used are considered safe and the equipment is generally less expensive than that of other imaging technologies [2].

On the other hand, several limitations should be mentioned, such as technical limitations (shadowing, reverberation, clutter artifacts) of the ultrasound. Similarly, tissue attenuation decreases ultrasound signal as a function of depth, limiting accurate assessment of deeper tissues, which is exacerbated by the presence of subcutaneous tissue [3]. Selection of the region of interest is also operator dependent and can be the reason for variability. In addition, methods that utilize external stimuli are highly subjective [3]. Along with the operator, an additional limitation also lies in the subject with their habitus and cooperation, which can be insufficient in children [4].

The inadequate standardization of the method and therefore its high variability of measurement results using different devices and probes is another limitation [4]. Also, potential risks arising from energetic ultrasonic push pulse in elastography increasing the thermal and mechanical index should be considered, especially in the case of vulnerable tissue [4].

The main limitation in UE research is inadequate standardization leading to incomparable studies and inconclusive results. Consequently, its use in clinical practice is limited. Current knowledge, presented through our article, is presented in Table 2, emphasizing clinical applications to several tissues along with their specifics (either advantageous (+) or disadvantageous (−)), if the data are available. General limitations and advantages should also be considered when performing UE.

## 17. Conclusions

Ultrasound elastography enhances the diagnostic and monitoring capability of traditional ultrasound, which is particularly advantageous in pediatrics, being safe, radiation-free and negating the need for sedation or general anesthesia during the imaging evaluation. It could therefore be easily introduced into everyday practice in several fields in pediatrics, especially in liver examination. In addition, other organs and tissues can be examined and evaluated using ultrasound elastography, presenting the urgent need for further research to determine clinical situations where the technique might provide some additional and important clinical information along with appropriate standardization of the method.

The aim of our study was to present the current status of UE use in children, which was accomplished and summarized; however, with increasing research and new knowledge obtained, further follow-up is needed.

## Figures and Tables

**Figure 1 children-10-01296-f001:**
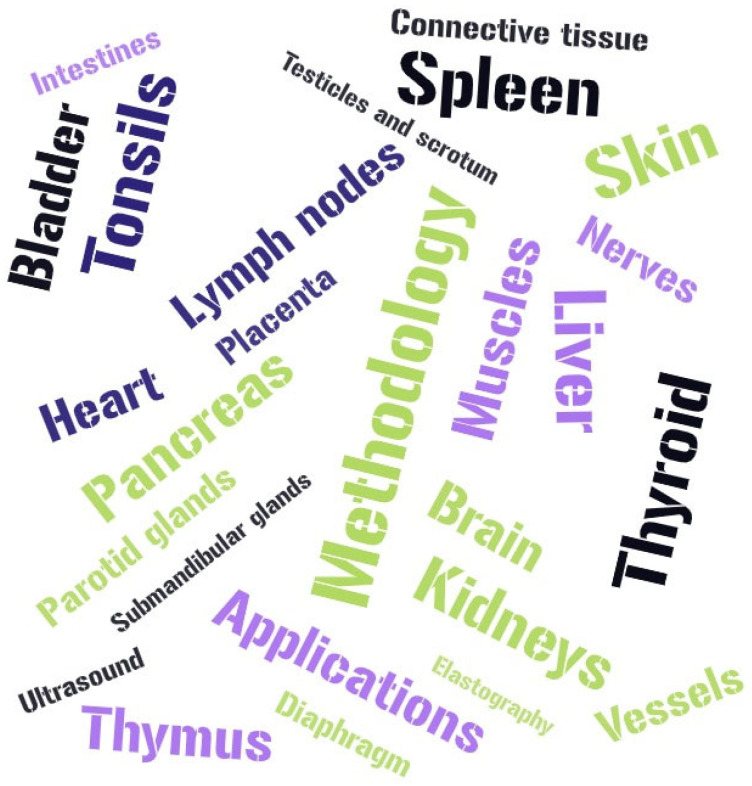
Reported topics of ultrasound elastography use and research in children.

**Figure 2 children-10-01296-f002:**
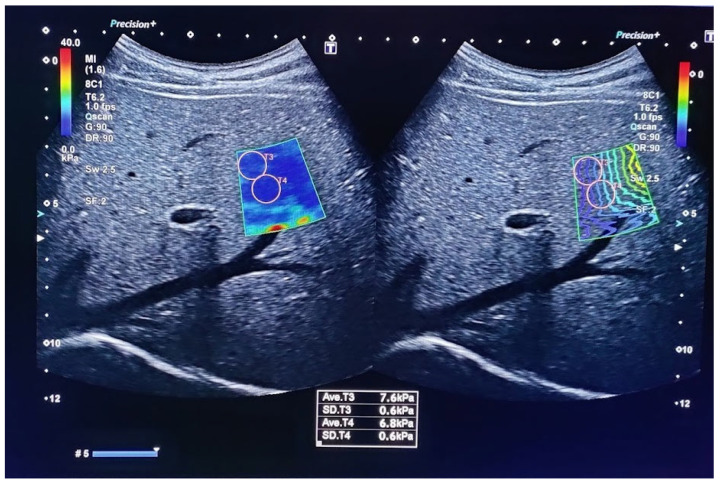
Ultrasound elastography using Canon aplio-a with maintenance of quality control by obtaining a standard deviation of less than 20% of the measurement as well as by maintaining an IQR/median ratio <30%.

**Figure 3 children-10-01296-f003:**
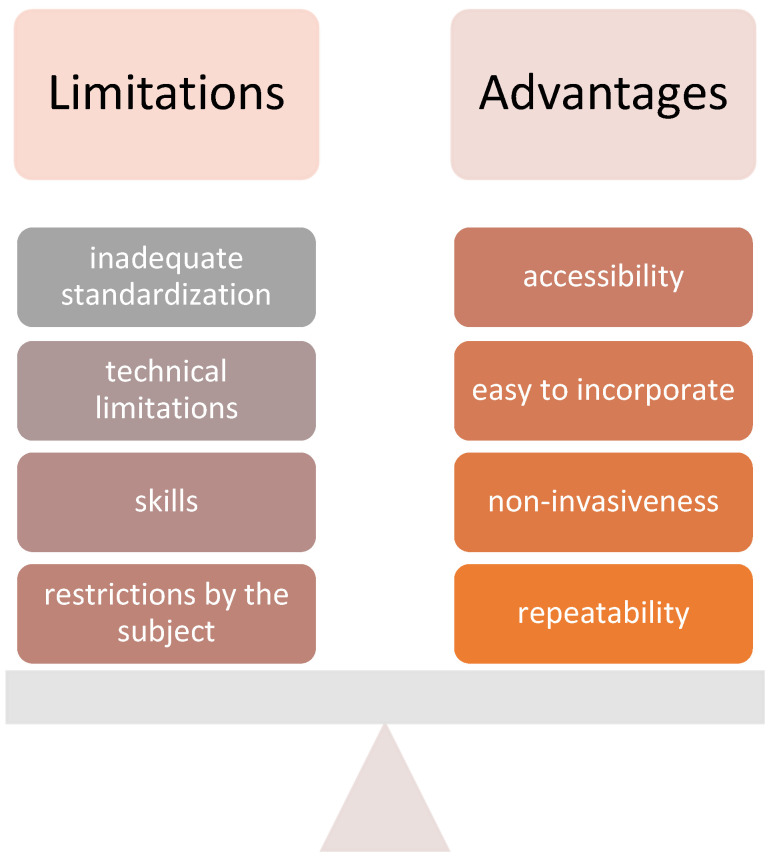
Main limitations and advantages of ultrasound elastography.

**Table 1 children-10-01296-t001:** Elastography techniques.

Elastography Technique	Description
Quasi-static method (strain imaging)	Based on a quasi-static deformation: a compression is applied to the tissue and the reference and the compressed images are compared
Vibro-acoustography (strain imaging)	Based on the ultrasound radiation pressure, detection of the movement of the ultrasound beam with a hydrophone
Point shear ware elastography (shear wave imaging)	Based on one focalized ultrasound beam with acoustic radiation force slightly displacing the tissue and then detecting the displacement
1D transient elastography (shear wave imaging)	Generation of a transient mechanical impulse and recording the shear wave that propagates within the tissue
2D shear wave elastography (shear wave imaging)	Extension of 1D elastography creating elasticity maps of biological tissues

**Table 2 children-10-01296-t002:** Clinical indications for ultrasound elastography in children; + advantage, − limitation.

**Tissue**	**Applications/Indications**	**Specifics**
Liver	Fibrosis assessment instead of liver biopsy regardless of the etiology of chronic liver disease	+Good liver fibrosis detection +Consistent liver stiffness values −Inconclusive liver fibrosis staging
Spleen	Splenomegaly, portal hypertension (stratification of the risk of having clinically significant varices)	−Inconclusive normal values
Pancreas	Type 1 diabetes mellitus (early detection, follow-up), cystic fibrosis	−Inconsistent results
Kidneys	Kidney elasticity in chronic kidney disease of various etiologies	+Progression of chronic kidney disease associated with ultrasound elastography results −Results dependent on the etiology −No association with histological samples
Bladder	Elasticity of bladder wall in neurogenic bladder	−Lack of studies
Thyroid	Malignancy differentiation, thyroid nodule biopsy (decision making), thyroiditis evaluation	−Inconsistent results −Lack of studies
Lymphatic tissue	Malignancy differentiation	−Cannot replace surgical biopsy −Lack of studies
Muscles	Cerebral palsy, scoliosis	+Monitoring in accordance with (physical) therapy −Lack of studies
Salivary glands	Malignancy differentiation	−Inconsistent results −Lack of studies
Brain	Marker of neurodevelopmental outcome	+Accessibility in babies −Vulnerable tissue −Lack of studies
Scrotum/testicles	Testicular torsion, space-occupying lesion, epididymal lesions	+Easily accessible +Monitoring children with hematological malignancies −Vulnerable tissue −Lack of studies
Skin	Scleroderma, burns	+Differentiation between normal and affected skin −Lack of studies

## Data Availability

Not applicable.

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
