# Peer review of "Ultrasound Elastography in Children"

_children, 2023, doi:10.3390/children10081296_

Round 1

Reviewer 1 Report

This review highlights the role of the US elastography in paediatric population, where ultrasound is a dominant imaging tool.

This topic is really interesting. However, I suggest to introduce a table that summarize the use of UE with the main advantage and limitation. Morever, I suggest to add more figures and I suggest a revision of table 1 because is not well structured. 

This sentence is not related with your work "Pancreatic inflammation also affects pancreas stiffness; however, studies are limited 182 to adult population [58,60]" I suggest to delete it.

Reviewer 2 Report

I would like to thank the author(s) of the paper for their contributions to knowledge expansion. To enhance the quality of the paper, I would like to offer the following additional suggestions:

1/ Reorganizing the introduction of the study and limiting it to three paragraphs in order to achieve the following goals:

- The significance of the current research is emphasized in the first paragraph.

- The second paragraph identifies the knowledge deficit that the current investigation aims to address.

- The third paragraph clarifies the research issue and how to approach it within the context of the purpose or objectives of the current study.

2/ Does the image in the article originate from the authors' work or from another source? Please respond to this query so that, if this image was obtained from another source, property rights can be taken into account to prevent future disputes in this context.

3/ The conclusion of the study is too lengthy and requires additional work that takes into account the abbreviation without bias, along with an explanation of whether or not the research problem has been solved, i.e., whether or not the current study accomplished its goal or objectives.

4/ Some references in the references section must be updated using references from 2023 and five years prior, along with the removal of all non-recent and unnecessary references.

//Good Luck//

This article required a minor English language revision in addition to the correction of a few typographical errors.
